# Synergistic Modification of Polyformaldehyde by Biobased Calcium Magnesium Bi-Ionic Melamine Phytate with Intumescent Flame Retardant

**DOI:** 10.3390/polym16050614

**Published:** 2024-02-23

**Authors:** Shike Lu, Xueting Chen, Bin Zhang, Zhehong Lu, Wei Jiang, Xiaomin Fang, Jiantong Li, Baoying Liu, Tao Ding, Yuanqing Xu

**Affiliations:** 1Henan Engineering Research Center of Functional Materials and Catalytic Reaction, Henan University, Kaifeng 475001, China; lushike1025@163.com (S.L.); liubaoying666@163.com (B.L.); dingtao@henu.edu.cn (T.D.); xuyuanqing@henu.edu.cn (Y.X.); 2College of Chemistry and Molecular Sciences, Henan University, Kaifeng 475001, China; cxt17861100039@163.com (X.C.); 13938523364@163.com (B.Z.); 3National Special Superfine Powder Engineering Research Center of China, Nanjing University of Science and Technology, Nanjing 210014, China; 15736870133@163.com (Z.L.); superfine_jw@126.com (W.J.)

**Keywords:** polyformaldehyde, phytate, synergistic effect, flame retardancy

## Abstract

Intumescent flame retardants (IFRs) are mainly composed of ammonium polyphosphate (APP), melamine (ME), and some macromolecular char-forming agents. The traditional IFR still has some defects in practical application, such as poor compatibility with the matrix and low flame-retardant efficiency. In order to explore the best balance between flame retardancy and mechanical properties of flame-retardant polyformaldehyde (POM) composite, a biobased calcium magnesium bi-ionic melamine phytate (DPM) synergist was prepared based on renewable biomass polyphosphate phytic acid (PA), and its synergistic system with IFRs was applied to an intumescent flame-retardant POM system. POM/IFR systems can only pass the V-1 grade of the vertical combustion test (UL-94) if they have a limited oxygen index (LOI) of only 48.5%. When part of an IFR was replaced by DPM, the flame retardancy of the composite was significantly improved, and the POM/IFR/4 wt%DPM system reached the V-0 grade of UL-94, and the LOI reached 59.1%. Compared with pure POM, the PkHRR and THR of the POM/IFR/4 wt%DPM system decreased by 61.5% and 51.2%, respectively. Compared with the POM/IFR system, the PkHRR and THR of the POM/IFR/4 wt%DPM system were decreased by 20.8% and 27.5%, respectively, and carbon residue was increased by 37.2%. The mechanical properties of the composite also showed a continuous upward trend with the increase in DPM introduction. It is shown that the introduction of DPM not only greatly reduces the heat release rate and heat release amount of the intumescent flame-retardant POM system, reducing the fire hazard, but it also effectively improves the compatibility between the filler and the matrix and improves the mechanical properties of the composite. It provides a new approach for developing a new single-component multifunctional flame retardant or synergist for intumescent flame-retardant POM systems.

## 1. Introduction

POM is a milky, opaque, crystalline, linear thermoplastic resin without a side chain whose structural formula is (CH_2_O)_n_. It has the advantages of good rigidity and hardness, excellent fatigue resistance, creep resistance, and chemical resistance, self-lubrication, a high thermal deformation temperature, stable mechanical properties, and good surface gloss, so it is widely used in electronic and electrical, light industry, machinery, building materials, and other fields [1]. However, because POM is flammable, there is a great fire hazard. Owing to its special chain molecular structure and ultra-high oxygen content, once ignited, it will continue to release a large amount of heat and toxic smoke, causing great harm to humans and the environment. Therefore, flame-retardant modification of POM is one of the difficulties and points of concern for many scientific researchers and related enterprises [2]. At present, adding flame retardants to polymers by blending is the simplest and most efficient way to alleviate this problem. Commonly used flame retardants include halogenated flame retardants, inorganic flame retardants, phosphorus and nitrogen flame retardants, silicon flame retardants, and intumescent flame retardants [3,4]. Intumescent flame retardants (IFR) have the advantages of being halogen-free, have low toxicity, are environmentally friendly, and are considered a class of flame retardants with the greatest application potential after halogen-based flame retardants [5,6,7]. However, IFR additives have poor compatibility with polymer matrices, low flame-retardant efficiency, and the additive amount is often above 30 wt% to achieve a certain flame-retardant effect, which makes the mechanical properties of the composite material greatly reduced [8].

According to the literature research, the commonly used strategies to solve the above problems mainly include the following: (1) search for novel efficient acid/char sources of IFR [9]; (2) surface modification of common IFR ingredient, such as APP [10]; (3) preparation of single-component multifunctional flame retardants [11]; (4) add synergistic agents [12,13,14], etc. Phytic acid (PA) is a natural organic phosphorus compound acid source extracted from plant seeds. It has a series of characteristics such as high storage capacity in nature, ultra-high phosphorus content, and excellent metal ion complexation ability, and it can react with organic amine compounds. These are great qualities for it to become a high-efficiency acid source in IFRs and to prepare single-component multifunctional flame retardants and organometallic salt synergists [15,16,17]. Cheng et al. used phytic acid as an acid source and chitosan and biochar as carbon sources to prepare flame-retardant coatings and applied them to the flame-retardant modification of cotton fabrics. The modified cotton fabric exhibited good thermal degradation and thermal oxidation stability, which reduces the fire hazard [9]. Yang et al. synthesized a new one-component intumescent flame retardant in the form of microporous nanosheets called hexa-(4-aminophenoxy) cyclotriphosphonitrile–phytic acid (HACP-PA), which assembled the carbon source, acid source, and gas source into a molecular structure. After adding 5 wt% HACP-PA to polylactic acid (PLA), the V-0 grade of UL-94 was achieved, and the LOI value was increased to 24.2%. Compared with PLA, the total heat release and peak heat release rate of the PLA composite with 5 wt% HACP-PA were reduced by 5% and 15.3%, respectively. In addition, the total smoke volume was also significantly reduced by 31.0%, and combustible volatiles were significantly reduced by the incorporation of HACP-PA [18]. Gong et al. added nickel phytate (PA-Ni) as a synergist with an intumescent flame retardant to modify polylactic acid (PLA). By adding 4 wt% PA-Ni and 11 wt% IFR, the resulting PLA composite can achieve the V-0 test grade of UL94. Compared with pure PLA, the peak heat release rate of PLA/11IFR/4PA-Ni is reduced by 62.3%, and the carbon residue is significantly increased [19]. Zhan et al. used layered melamine phytates (MEL-PA) with intumescent flame retardants in flame-retardant polypropylene (PP). Studies have shown that the addition of MEL-PA can effectively improve the limiting oxygen index of PP composites, greatly reduce the heat and smoke release of PP/IFR, and increase the degree of graphitization of the carbon layer, which proves that MEL-PA and IFRs have a good synergistic effect on flame-retardant PP [20].

In this paper, a biobased calcium magnesium bi-ionic melamine phytate (DPM) was designed and prepared. It was employed to modify POM by compounding with the IFR, containing ammonium polyphosphate (APP), benzoxazine (BOZ), and melamine (ME) as the acid source, carbon source, and gas source, respectively; it was fabricated and optimized in our previous works [19,21]. The flame retardancy and mechanical properties of the intumescent flame-retardant POM composites with DPM were studied. It was found that DPM not only has an excellent synergistic flame-retardant effect with IFRs but also enhances the compatibility between fillers and substrates, improves the mechanical properties of composite materials, and has good application prospects. It provides a new approach for the development of biobased environmentally friendly intumescent flame retardants.

## 2. Experimental Section

### 2.1. Materials

Phytic acid (PA, 70% aqueous solution), magnesium hydroxide (99.5%), and calcium hydroxide (99.5%) were purchased from Xiya Chemical Co., Ltd, Chengdu, China. 4, 4-diaminodiphenyl sulfone (DDS, 99.5%) was obtained from Shanghai Meryer Technologies Co., Ltd., Shanghai, China. POM (POM, MC90), melamine (ME, 99%), and antioxidant (1010, industrial grade) were provided by Henan Kaifeng Longyu Chemical Co., Ltd, Kaifeng, China. APP (TF-201, crystal form II, *n* ≥ 1000) was purchased from Shifang Taifeng New Flame Retardant Co., Ltd, Shifang, China. Benzoxazine (BOZ) was made in our laboratory [22].

### 2.2. Preparation of DPM

Firstly, 1.5 equivalent of calcium hydroxide and 1.5 equivalent of magnesium hydroxide were dispersed in deionized water (0.2 mol/L) at room temperature to prepare a calcium magnesium bimetallic ion suspension solution, and 1 equivalent of phytic acid and 4 equivalent of melamine were dissolved in deionized water separately to prepare a phytic acid solution (0.2 mol/L) and melamine suspension (0.2 mol/L). Then, 1 equivalent of DDS was dissolved in ethanol (0.5 mol/L) for use. The preparation diagram is shown in Figure 1.

Secondly, the DDS ethanol solution was slowly and uniformly added to the phytic acid aqueous solution at room temperature. After stirring for 30 min, the calcium and magnesium bimetallic ion aqueous solution was added slowly and evenly to the above reaction solution. Then, it was stirred for another 30 min, the temperature was raised to 85 °C, and the melamine suspension was added dropwise. A large amount of precipitation was quickly generated, and the temperature was kept with stirring for 4 h until the reaction was completed.

Finally, DPM was obtained by filtering, washing, and drying, with a yield of 92.0%.

### 2.3. Preparation of POM Composites

POM pellets, intumescent flame retardant (IFR) (composed of APP/BOZ/ME), and DPM were dried at 80 °C in an electric blast drying oven for more than 4 h. After pre-mixing all the ingredients with weighted amounts according to the design formula, the mixture was melt-blended, extruded, and granulated by a twin-screw extruder at 165 °C to 175 °C. The obtained pellets were further dried at 80 °C for more than 6 h. After thorough drying, they were injected by an injection molding machine to make standard bars for further testing. The plasticizing temperature of the injection molding machine was 170 °C to 180 °C.

### 2.4. Characterization

Fourier transform infrared spectroscopy (FTIR) analysis was performed by the Nicolet 170sx Fourier Infrared spectrometer (Bruker Spectrometer, Saarbrücken, Germany), using potassium bromide tablets with an optical test range of 400 cm^−1^ to 4000 cm^−1^ and a resolution of 4 cm^−1^.

The vertical combustion performance (UL-94) was tested on the horizontal vertical combustion tester (Suzhou Testech Testing Instrument Technology Co., Ltd., Suzhou, China) according to the GB/T 2408-2008 [23] test standard, with a spline size of 125 × 12.5 × 3.2 mm.

The limiting oxygen index (LOI) was tested in accordance with GB/T 2406.2-2009 [24] standard on the smart oxygen index fume hood integrated machine (Suzhou Testech Testing Instrument Technology Co., Ltd., Suzhou, China), and the sample size was 80 × 10 × 4 mm.

The cone calorimetry (Fire Testing Technology Ltd., East Grinstead, UK) was measured using the ISO 5660-1 2015 [25] standard to study the combustion behavior at a thermal radiant flux of 50 kw/m^2^ and a temperature of 700 °C.

The thermal degradation gas infrared analysis test was carried out in a combination of a thermogravimetric analyzer (TGA/SDTA 851, Mettler-Toledo, Zurich, Switzerland) and Fourier infrared tester (INVENIOS, Bruker, Saarbrücken, Germany), heated from room temperature to 800 °C in a nitrogen atmosphere at a heating rate of 10 °C/min and a gas flow rate of 50 mL/min.

The surface topography of the coke slag after cone calorimetry was analyzed by scanning electron microscopy (SEM, JEOL JSM-7610F, UK). The test voltage was 5 kV. The scanning electronic microscope was equipped with an energy-dispersive X-ray spectrometer (EDS) for elemental analysis using elemental mapping with a voltage of 8 kV and a detection limit of 0.01%.

Elemental analysis was performed by X-ray fluorescence spectroscopy (XRF, S2 RANGER, Bruker) with a test range of 11Na-92U and a content range of ppm-100%.

The degree of graphitization of the carbon layer was measured by a laser microscope Raman spectrometer (Renishaw in Via, Renishaw, London, UK). The excitation wavelength was 532 nm and the spectral range was 1000–2000 cm^−1^.

The mechanical properties of the samples were tested by an electronic universal testing machine (TCS-2000, GOTECH Testing Machines Inc., Taiwan) at room temperature. The tensile test was conducted according to the GB/T 1040.1-2006 [26] standard, the tensile rate was 50 mm/min, and the sample size was 150 × 10 × 4 mm. The bending test was conducted according to the GB/T 9341-2008 [27] standard, the drop rate was 2 mm/min, and the sample size was 80 × 10 × 4 mm.

According to the standard GB/T 1043-2008 [28], a sample with a size of 80 × 10 × 4 mm was used to open a V-shaped notch with a depth of 2 mm in the center, and the notch impact test was carried out with the impact testing machine (ZBC-8400-C, GOTECH Testing Machines Inc., Taiwan).

## 3. Results and Discussion

### 3.1. Characterization of DPM

The FTIR spectra of DPM are shown in Figure 2. According to the spectrum, DPM retained a -CH vibration absorption peak near 2900 cm^−1^, and O=P and O-P characteristic absorption peaks appeared near 1640 cm^−1^ and 1100 cm^−1^, respectively. It can be proved that DPM retains the basic structure of phytic acid [29]. The characteristic peaks of -NH_2_ and triazine rings in ME appear near 1510 cm^−1^ and 800 cm^−1^, and the characteristic absorption peaks of a benzene ring and -SO_2_- groups of DDS appear at 3200–3500 cm^−1^ and 1100 cm^−1^ wave numbers [30]. In addition, the characteristic peak of -OH near 3700 cm^−1^ was not clearly reflected in DPM, indicating that the successful grafting of ME and DDS and the full reaction of Mg(OH)_2_ and Ca(OH)_2_ as raw materials had been achieved.

Further combined with EDS map test analysis (Figure 3), the atomic percentages of P elements and S, Ca, and Mg elements in DPM are 3.18%, 0.37%, 0.75%, and 0.89%, respectively, and the ratio between these four major elements is 1:0.12:0.24:0.28, which is roughly similar to the theoretical value (1:0.17:0.25:0.25), and it can be seen from the figure that the distribution of elements in DPM is very uniform. Finally, according to the results of XRF analysis (Table 1), it can be seen that the ratio of major elements P, S, Ca, and Mg in DPM is 1:0.29:0.41:0.29, which is roughly consistent with the theoretical value (1:0.17:0.25:0.25). Combined with FITR, EDS maps, and XRF analysis results, it can be inferred that DPM was successfully prepared.

Figure 4 shows the TG and DTG diagrams of DPM. As can be seen from the figure, the initial decomposition temperature of DPM (T_−5%_) is 229.7 °C, the main maximum weight loss rate temperature (T_max_) has two intervals, T_max1_ and T_max2_, which are 305.2 °C and 537.2 °C, respectively, and the residual amount at 800 °C is 43.6%.

Figure 5 shows the real-time infrared spectrum of gas released by DPM under nitrogen thermogravimetric conditions. It can be seen from the three-dimensional figure that under the maximum decomposition rate temperature (305.2 °C) (Figure 5), the asymmetric stretching vibration peak and bending vibration peak of carbon dioxide appear near 2349 cm^−1^ and 667 cm^−1^. N-H bending vibration absorption peaks emerge near 1630 cm^−1^ and 950 cm^−1^. There are P-O stretching vibrations and bending vibrations in the vicinity of 1600–1740 cm^−1^. The characteristic absorption peak of the sulfonic acid group was found near 1000–1300 cm^−1^. In addition, at 143.7 °C, the peak value near 950 cm^−1^ is higher, and the corresponding NH_3_ or other amino volatiles may be generated from the early release of partially exposed amino groups in the DPM structure. The results showed that CO_2_, a small amount of amines, a very small quantity of phosphate–oxygen radicals, and sulfonic acid groups were released by thermal decomposition of DPM at this temperature.

### 3.2. Flame-Retardancy Analysis

Vertical combustion (UL-94) and limited oxygen index (LOI) tests are the basic and intuitive ways to observe the flame-retardant properties of polymeric materials. The UL-94 grade is mainly judged according to the total burning time (Tall) after two ignites. The shorter the T_all_, the more difficult the material is to ignite. The LOI indicates the minimum oxygen concentration that can support the combustion of the material, and the higher the LOI, the better the flame-retardant performance.

Table 2 shows the UL-94 grade and LOI values of the DPM and IFR synergistically flame-retardant POM composites. Pure POM is extremely flammable, and the combustion process is accompanied by a large number of droplets; the UL-94 test cannot pass any grade, and the LOI is only 15%. After the addition of the IFR, the UL-94 test time of the POM/IFR system has been significantly reduced; its UL-94 reached V-1 grade, and the LOI has been significantly increased to 48.5%. When the amount of DPM is increased from 1% to 8%, the combustion performance of each system is further improved. The test results show that with the introduction of DPM synergism, the UL-94 and LOI of POM composites show a trend of first increase and then decrease. The UL-94 of the POM/IFR/4 wt%DPM and POM/IFR/5 wt%DPM systems can reach the V-0 grade with an LOI of 59.1% and 57.6%, respectively. In contrast, the POM/IFR/4 wt%DPM system shows the best flame retardancy.

According to the results in Table 2, we subsequently selected four representative systems, namely pure POM, POM/IFR, POM/IFR/1 wt%DPM, and POM/IFR/4 wt%DPM, to further investigate their combustion behavior and mechanism of action.

### 3.3. Investigation of Combustion Behavior

Cone calorimetry (CONE) simulates real fire conditions, giving us a lot of information about the combustion process. Due to the comprehensiveness and reliability of the data, it has been widely used in the flame-retardancy evaluation of various composite materials, such as various engineering plastics, foam materials, and wood materials. The characteristic data of ignition time (TTI), heat release rate (HRR), total heat release (THR), effective heat of combustion (EHC), total smoke release (TSP), and specific extinction area (SEA) can be obtained [31,32].

Figure 6 shows the curves of HHR, THR, smoke release rate (SPR), and TSP of the POM composites, respectively, and the specific data are shown in Table 3. From the test data, pure POM is extremely flammable, the ignition time (TTI) is 43 s, and after ignition, it burns violently to completion without any residue. The peak heat release rate (PkHRR) is 335.55 kW/m^2^, and the average heat release amount (AvHRR) is 233.15 kW/m^2^. The THR was 133.08 MJ/m^2^, the mean effective heat of combustion (MeanEHC) was 14.54 MJ/kg, the mean mass loss rate (AvMLR) was 19.71 g/(m^2^·s), and the SEA, TSP, and carbon residue were all zero. All composites with added flame retardants have shorter TTI than pure POM, which is caused by the premature decomposition of the IFR before the POM matrix at high temperatures. The PkHRR, AvHRR, and THR of POM/IFR composites with 30%IFR were reduced to 163.17 kW/m^2^, 42.49 kW/m^2^, 89.47 MJ/m^2^, with a MeanEHC of 10.51 MJ/kg and an AvMLR of 4.13 g/(m^2^·s). Compared with pure POM, SEA, TSP, and carbon residue increased to 86.30 m^2^/kg, 7.57 m^2^ and 11.3%, respectively.

When 1% DPM was introduced, the PkHRR, AvHRR, THR, MeanEHC, and AvMLR of the composite were further reduced, and the SEA, TSP, and carbon residues were increased to a certain extent. With the increase in DPM introduction, the test results of the composite materials were improved. The PkHRR, AvHRR, THR, MeanEHC, and AvMLR of the POM/IFR/4 wt%DPM system were further reduced to 129.17 kW/m^2^, 32.71 kW/m^2^, 64.90 MJ/m^2^, 8.33 MJ/kg, and 3.7 g/(m^2^·s). Compared with pure POM and POM/IFR systems, the THR of the POM/IFR/4 wt%DPM system was reduced by 51.2% and 27.5%, respectively, and the SEA and TSP were also decreased to 109.31 m^2^/kg and 6.8 m^2^, respectively; the carbon residue was increased to 15.5%. The above data all show that the introduction of DPM greatly improves the flame-retardant efficiency of the IFR, significantly reduces the amount of heat released during combustion, improves the amount of carbon residue of composite materials, and strengthens the flame-retardant effect of the condensed phase. At the same time, it also inhibits the combustion of the material in the gas phase, resulting in the release of a large amount of refractory smoke during the combustion process, which plays the role of reducing the amount of oxygen, combustible gas, and heat released during the combustion process. These results indicate that DPM does not only have a flame-retardant effect in the gas phase but also promotes the condensed phase flame-retardant effect of the crosslinked carbon of flame-retardant POM composites and is a good flame-retardant synergist for the intumescent flame-retardant POM system.

### 3.4. Carbon Residue Analysis

The digital photos, SEM, and LRS of residual carbon after the combustion of flame-retardant POM composites are shown in Figure 7. After pure POM is ignited, it burns violently until complete without any carbon residue generation. From the digital photos, the carbon residue of the POM/IFR system has increased significantly, reaching 11.3%, but the surface carbon layer is relatively loose, fragmentary, soft, and without any luster. After the introduction of DPM, the amount of residual carbon in the system is further increased; the POM/IFR/1 wt%DPM system reaches 15.0%, the POM/IFR/4 wt%DPM system reaches 15.5%, and the quality of the carbon layer is obviously improved, is more compact, and the surface shows a certain metallic luster. In particular, the carbon layer generated by the POM/IFR/4 wt%DPM system after combustion appears as a whole, and the external carbon layer basically has no damaged carbon residue, almost completely covering and protecting the internal carbon layer, and the amount of carbon residue has further increased. Thus, the carbon layer shows excellent quality and can play an efficient barrier role.

It can be seen more obviously from the SEM diagram of carbon residue (Figure 7) that the surface of the carbon layer of the POM/IFR system is broken and there are many carbon slag and holes on the surface that cannot effectively block the transfer of oxygen, heat, and combustible gas mixture, so the POM/IFR system does not achieve the ideal flame-retardant effect. After the introduction of DPM, the carbon layer of the POM/IFR/1 wt%DPM system is compact and continuous, showing a certain metallic luster, but there are still a few holes and broken carbon residue. The POM/IFR/4 wt%DPM system has a good improvement, and the carbon layer is completely and continuously arranged in a compact and dense manner, basically without broken carbon residue and holes, which can exert effective heat insulation and oxygen insulation, block the diffusion and transfer of combustible gas mixture during combustion, and play a good shielding and protecting role in the interior of the system. Therefore, the POM/IFR/4 wt%DPM system shows outstanding flame retardancy.

In order to further analyze the carbon layer quality, the carbon layer after the CONE test was analyzed by Raman spectroscopy (LRS) test, as shown in Figure 7. The LRS of the carbon layer mainly has two peaks, one near 1350 cm^−1^ (peak D) and the other near 1580 cm^−1^ (peak G), which represents sp^2^ hybridization of the ordered carbon structure with in-plane stretching vibrations [33,34]. The graphitization degree of the carbon layer is usually expressed by the peak area ratio (I_D_/I_G_) of peak D to peak G; the smaller the I_D_/I_G_ value, the higher the degree of graphitization and the better the quality of the carbon layer. The I_D_/I_G_ values of carbon residue in the POM/IFR, POM/IFR/1 wt%DPM, and POM/IFR/4 wt%DPM systems were 1.35, 0.99, and 0.87, respectively. The graphitization degree of the carbon layer in the POM/IFR/4 wt%DPM system was the highest. The results show that the introduction of DPM in the flame-retardant POM can effectively improve the degree of graphitization of the carbon layer, obtain a higher quality carbon layer, effectively provide heat insulation, and protect the interior of the composite material, resulting in excellent flame retardancy of the composite material. The above analysis shows that the comprehensive flame-retardant effect of the POM/IFR/4 wt%DPM system is the best, which is basically consistent with the combustion test data (UL-94 reaches V-0 grade, LOI up to 59.1%).

### 3.5. Thermogravimetric Infrared Analysis

Figure 8 shows the thermogravimetric analysis (TG and DTG) curves of POM and flame-retardant POM composites under a nitrogen atmosphere, and the specific data are shown in Table 4. The initial decomposition temperature T_−5%_ of pure POM is 300.7 °C, the maximum decomposition rate temperature T_max_ is 338.8 °C, and no carbon residue was generated at 600 °C. Compared with POM, the T_−5%_ of the POM/IFR system decreased significantly to 261.7 °C, which was caused by the advanced decomposition due to the introduction of the IFR; T_max_ was 269.1 °C, which was also significantly advanced, and the carbon residue increased to 17% at 600 °C. The results show that although the introduction of the IFR can reduce the thermal stability of the composite, it can effectively improve the carbon formation property of the composite. After the introduction of DPM, the T_−5%_ and T_max_ of the POM/IFR/1 wt%DPM and POM/IFR/4 wt%DPM systems are basically the same as those of POM/IFR systems, except that their residual carbon content increases to 21.8% and 23.5%, respectively, at 600°C, both of which are higher than the theoretical carbon residue content of their respective systems. It can be shown that the introduction of DPM does not play a single role but has an excellent synergistic effect with the IFR, which can more efficiently promote the carbon formation of the flame-retardant POM composite.

Figure 9 shows the thermogravimetric infrared (TG-IR) analysis results of pure POM and flame-retardant POM composites. As can be seen from the three-dimensional figure, the gas products during the combustion of POM and the flame-retardant POM system are basically similar. The temperature range of pure POM gas release is about 250~375 °C, while the temperature range of the POM/IFR and POM/IFR/4 wt%DPM systems is greatly shortened to around 240~260 °C and 245~265 °C, respectively. According to the FTIR spectra (Figure 10), the gas products of pure POM at the maximum decomposition rate are mainly carbonyl, carbonyl compounds (1715 cm^−1^–1770 cm^−1^), and hydrocarbon compounds (2640 cm^−1^–2940 cm^−1^). However, with the introduction of the IFR and DPM, the gas products of the POM/IFR and POM/IFR/4 wt%DPM systems show weak characteristic absorption peaks near 3500 cm^−1^; this may be the water released by the intumescent flame-retardant POM system during the carbonization crosslinking reaction under high-temperature conditions, which can dilute the combustible volatile matter and oxygen concentration to a certain extent and improve the flame retardancy of the composite material. Compared with the POM/IFR system, there is no obvious difference in the gas generated by the POM/IFR/4 wt%DPM system. Combined with the cone calorimetric analysis of each system and the carbon residue analysis, it can be seen that mutual reactions occurred among DPM, IFR, and POM in the combustion process. DPM mainly plays a catalytic role in the condensed phase, leading to rapid carbonization and more high-quality char residue [29]. The ME grafted on phytate still retains some -NH_2_ groups, and some of the amine refractory gas mixture generated during combustion is further oxidized into nitrogen oxide or nitrogen and water at high temperatures, thus diluting the concentration of combustible volatiles in the combustion zone and playing a certain flame-retardant role in the gas phase [30].

### 3.6. Flame-Retardant Mechanism Analysis

The mechanism of the DPM-synergistic IFR to flame-retardant POM is shown in Figure 11. First, at the initial stage of combustion, the acidic substance released by the thermal decomposition of APP reacts with the charring agent BOZ for esterification and promotes the ring-opening and crosslinking of BOZ to interweave with the polymer matrix to form a three-dimensional network carbon layer [33]. Ca^2+^ and Mg^2+^ in DPM can not only catalyze the esterification reaction between the acid and carbon source but also strengthen the carbon layer, which is consistent with the flame-retardant mechanism of phytic acid metal salts reported in the literature [29]. Secondly, DPM can also release a certain amount of amino compounds, ·SO, ·PO_2_, and ·PO free radicals when heated to capture highly active free radicals maintaining combustion such as ·H and ·OH in the gas phase. Finally, when ME is decomposed by heat, a large amount of water vapor, CO_2_, and other nonflammable gases are generated [30]. While diluting the concentration of flammable gas and oxygen, the molten carbon layer is rapidly intumescent and foamed, forming a dense intumescent carbon foam layer, which has a shielding effect and thus achieves a good flame-retardant effect. DPM has an excellent synergistic effect with the IFR. The introduction of it can strengthen the quality of the carbon layer and improve its shielding effect. In addition, when heated, DPM will release refractory gas and free radical catchers to strengthen the dilution effect and quenching effect. Therefore, DPM can be used as a good flame-retardant synergist for the intumescent flame-retardant POM system.

### 3.7. Mechanical Property Analysis

Table 5 shows the test results of mechanical properties of pure POM and flame-retardant POM composites. From the test data, pure POM has excellent mechanical properties. However, after the introduction of the IFR, the mechanical properties of the POM/IFR system deteriorated sharply. The notch impact strength, bending strength, and tensile strength decreased to 2.78 kJ/m^2^, 48.04 MPa, and 36.20 MPa, respectively, which decreased by 50.5%, 31.7%, and 42.9% compared with pure POM. The compatibility between the IFR and the matrix is not good. When DPM is introduced into the intumescent flame-retardant POM system, the mechanical properties of the composite are comprehensively improved, showing a continuous upward trend with the increase in DPM addition. The notch impact strength, bending strength, and tensile strength of POM/IFR/4 wt%DPM reached 3.28 kJ/m^2^, 51.24 MPa, and 38.68 MPa, respectively; of note, the notch impact strength increased by 18.0% compared with POM/IFR, with excellent flame retardancy and good mechanical properties.

In order to further explore the reasons why DPM improves the mechanical properties of composite materials, the impact cross-sections of pure POM and flame-retardant POM composites were scanned by SEM, as shown in Figure 12. After the sections were enlarged 1000 times, it can be seen that the pure POM impact section is very rough, showing obvious ductile fracture traces without any longitudinal cracks and holes. After the addition of the IFR, the impact cross-section of the POM/IFR system presents a large height difference, which is full of particles with uneven particle size, and there are cracks and holes at the joints with the matrix, and the interface separation of the two phases is serious, which indicates the poor compatibility of the IFR and POM matrix, leading to serious deterioration of the mechanical properties of the POM/IFR composites. Compared with the POM/IFR system, the height difference of the impact section of the POM/IFR/4 wt%DPM system has been significantly reduced, and the surface is rougher, but there are still some additive particles with uneven particle size on the surface, and there are still small cracks and holes at the two joints. With the increase in DPM addition and the decrease in IFR addition, in the POM/IFR/8 wt%DPM system, the impact fracture and hole phenomenon improved, and the two joints are tighter without obvious cracks, which also makes the composite perform better in terms of mechanical properties. The improvement in the mechanical properties of flame-retardant POM composites by DPM may be due to the special structure of DPM as a chain macromolecular organometallic amine salt synergist, which improves the compatibility between POM and filler, contributes to the dispersion of the intumescent flame-retardant system during processing, and improves the interface interaction between the filler and the matrix.

## 4. Conclusions

In this paper, DPM was prepared and characterized. The effects of its synergy on the flame retardancy and mechanical properties of POM were discussed under the condition that the total additive amount of the IFR and DPM remained at 30 wt%. The results were as follows:

(1) Through the super-strong metal complexation ability of PA and the reaction with organic amine compounds, the cationic unsaturated state of calcium and magnesium bi-ionic melamine phytates were connected with diamine DDS to prepare DPM. The reaction conditions were simple and mild, and the yield was very high (92%).

(2) The use of DPM as a synergist in the intumescent flame-retardant POM system can effectively improve the flame retardancy of POM composite materials, especially the POM/IFR/4 wt%DPM system. Its UL-94 test passed the V-0 grade and the LOI reached the highest value, 59.1%, while the THR decreased by 51.2% compared with pure POM. Compared with the POM/IFR system, its THR decreased by 27.5%, and the carbon residue increased significantly to 15.5%. Moreover, the synergistic system of DPM and IFR cannot only improve the carbonization ability of the system, leading to the formation of a dense carbon layer, but it can also inhibit the combustion of the gas phase. The gas phase and the condensed phase cooperate to perform an efficient flame-retardant effect so that the flame retardancy of the DPM synergistic flame-retardant system is significantly improved.

(3) DPM as a synergist in the intumescent flame-retardant POM system can effectively improve the mechanical properties of the composite materials. Compared with the POM/IFR system, the mechanical properties of the composite materials basically show a continuously rising trend with the increase in DPM addition, and the SEM analysis of the section also proves that DPM can improve the interface interaction of POM and filler and enhance their compatibility.

## Figures and Tables

**Figure 1 polymers-16-00614-f001:**
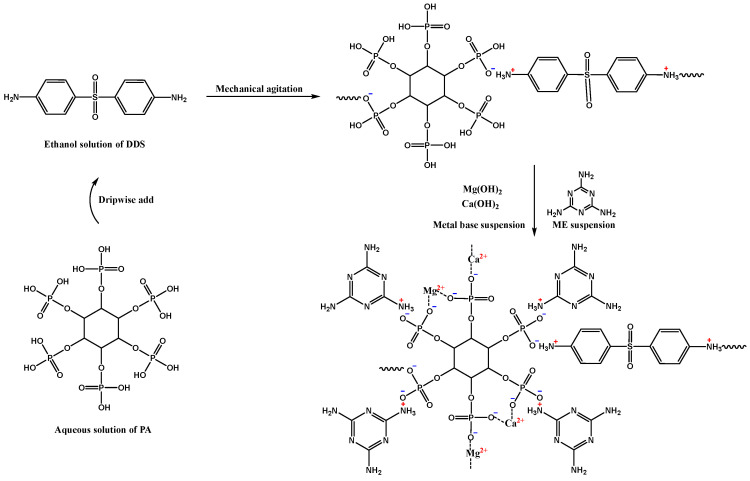
Preparation route and one of the possible structural schematic diagrams of DPM.

**Figure 2 polymers-16-00614-f002:**
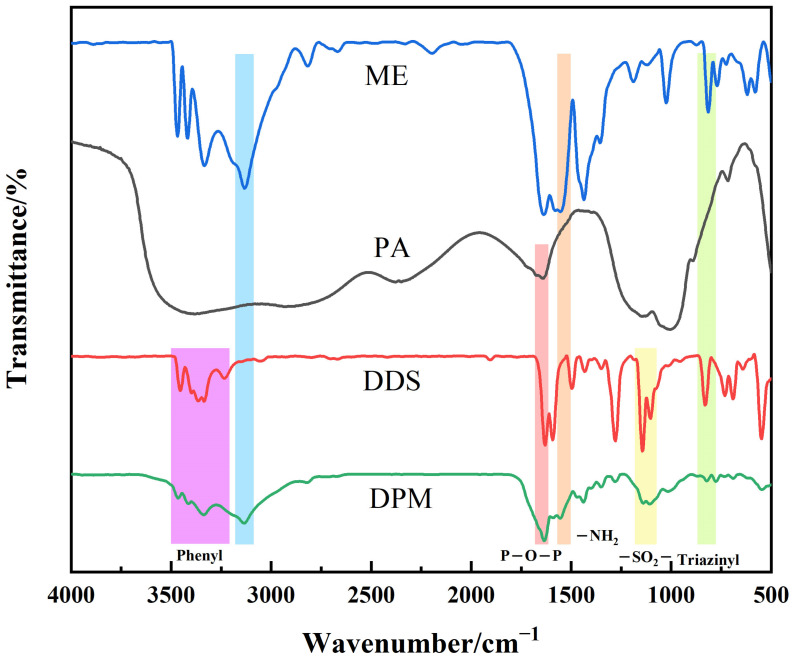
FTIR spectra of DPM.

**Figure 3 polymers-16-00614-f003:**
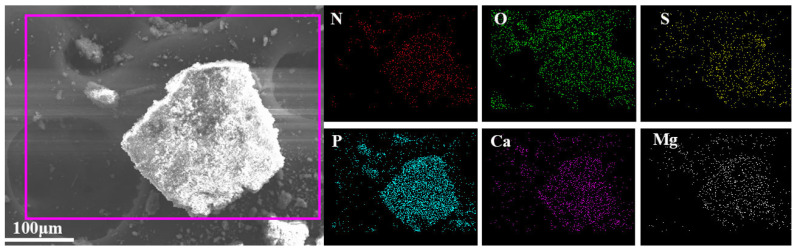
SEM images and EDS map results of DPM.

**Figure 4 polymers-16-00614-f004:**
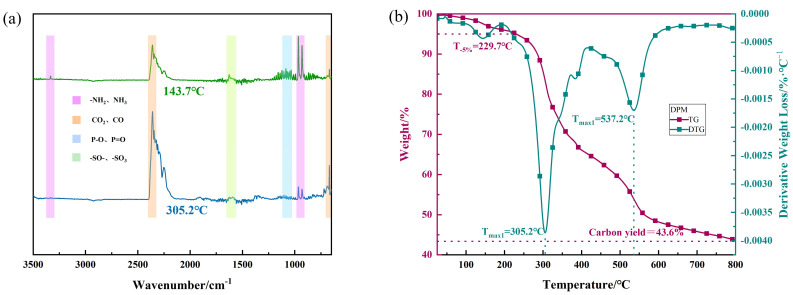
FTIR spectra of DPM thermal decomposition products at different temperatures (**a**) and thermogravimetric curves (**b**).

**Figure 5 polymers-16-00614-f005:**
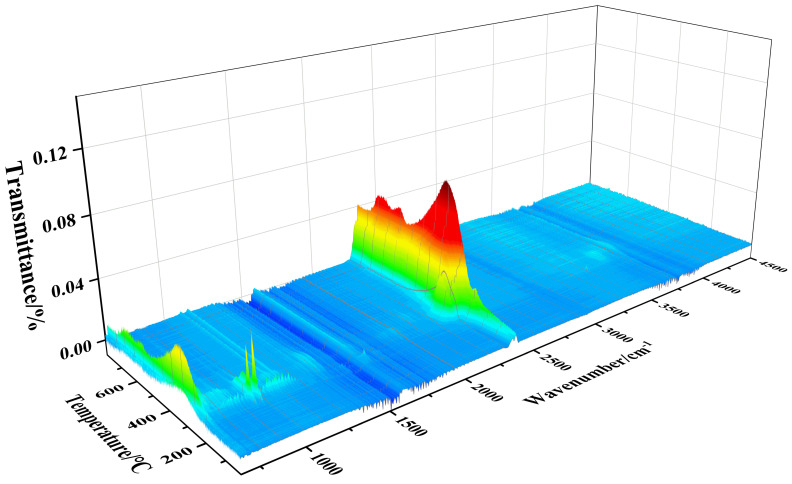
FTIR 3D diagram of thermal decomposition products of DPM at real-time temperature.

**Figure 6 polymers-16-00614-f006:**
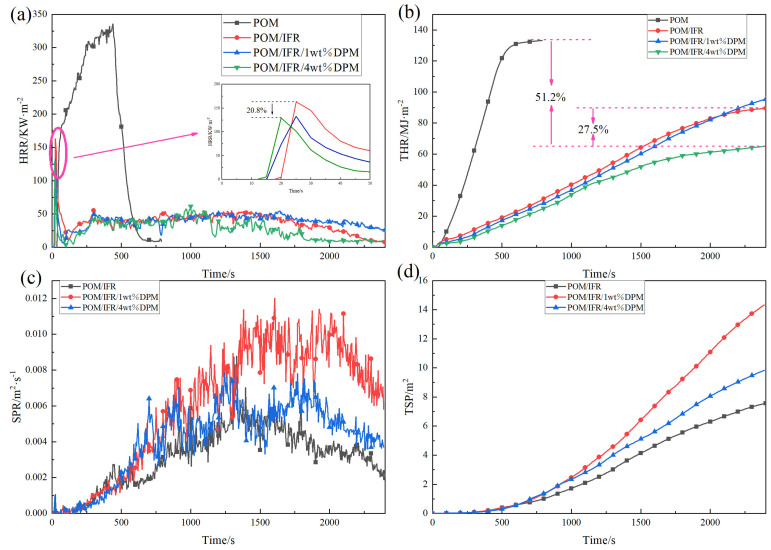
HRR curve (**a**), THR curve (**b**), SPR curve (**c**), and TSP curve (**d**) of composite material obtained by CONE test.

**Figure 7 polymers-16-00614-f007:**
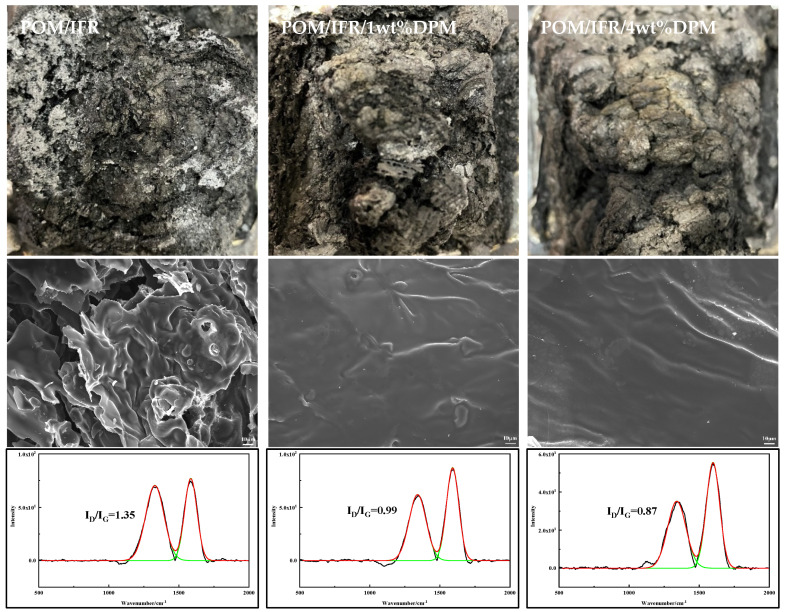
Digital photos, SEM images, and LRS maps of carbon residue after combustion of composite materials.

**Figure 8 polymers-16-00614-f008:**
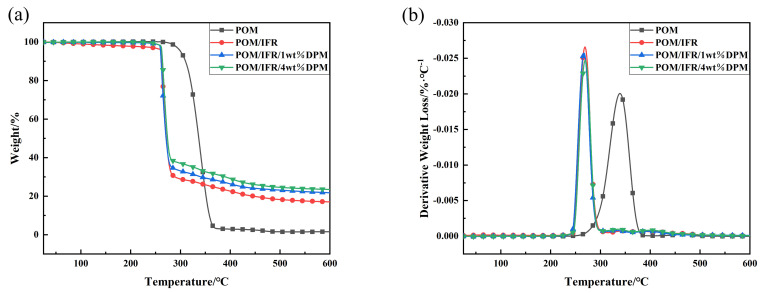
TG (**a**) and DTG (**b**) curves of composite materials.

**Figure 9 polymers-16-00614-f009:**
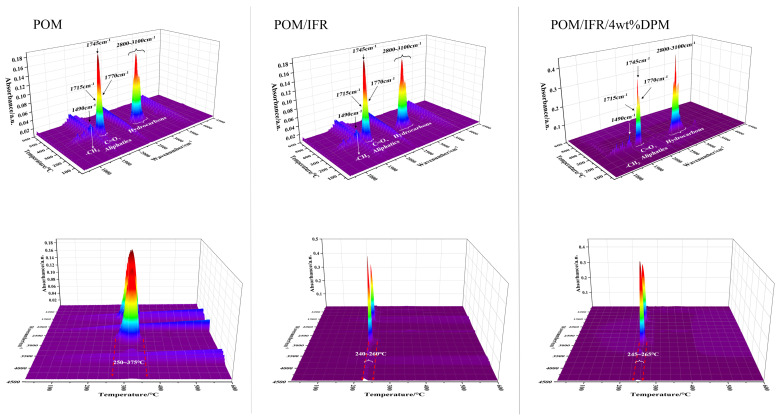
Real-time TG-IR 3D image of composite materials.

**Figure 10 polymers-16-00614-f010:**
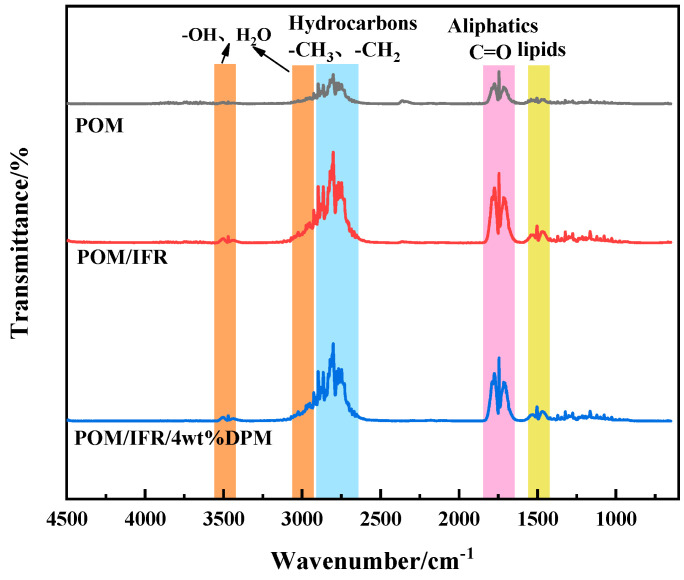
Infrared spectrum of gas released by composite at real maximum thermogravimetric rate.

**Figure 11 polymers-16-00614-f011:**
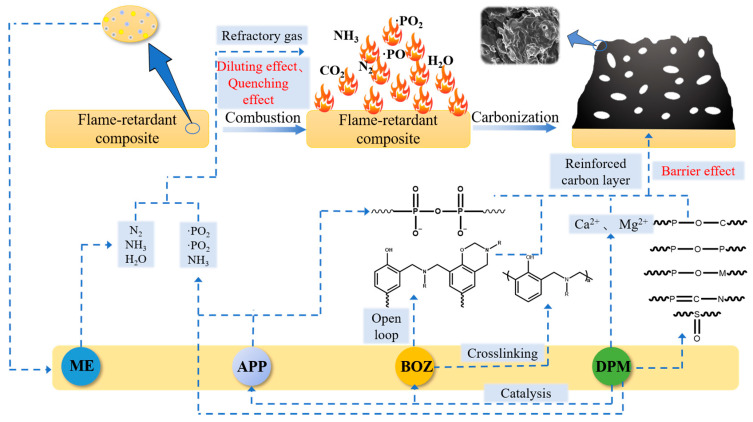
Flame-retarding mechanism diagram of DPM-synergistic IFR flame-retarding POM.

**Figure 12 polymers-16-00614-f012:**
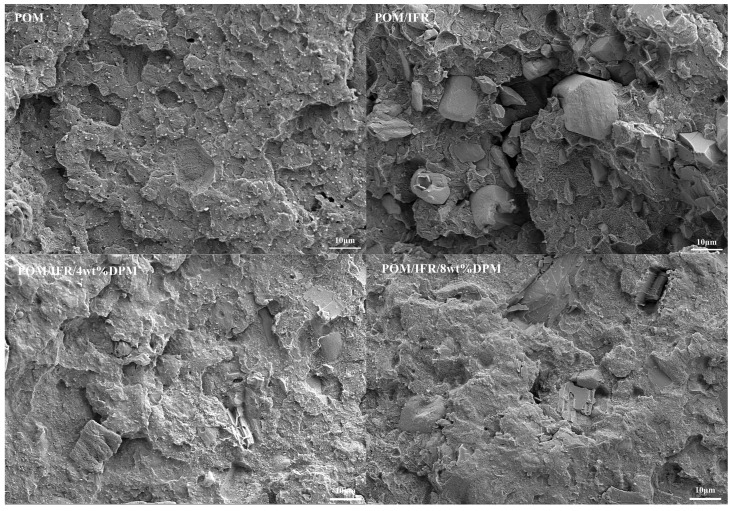
SEM image of composite section.

**Table 1 polymers-16-00614-t001:** Results of XRF analysis of DPM.

Component	Weight/%	Element	Atomic Ratio (Relative to the P Element) *
P_2_O_5_	55.09	P	1.00
SO_3_	17.88	S	0.29
CaO	17.64	Ca	0.41
MgO	9.10	Mg	0.29
Cl	0.17	--	--
Rest	0.16	--	--

Note: “*” represents the ratio of the atoms of each element to the P element in DPM.

**Table 2 polymers-16-00614-t002:** Flame-retardancy data of POM composites.

Sample	UL-94 (3.2 mm)					LOI/%
	t_1_/s	t_2_/s	T_All_/s	Dripping	Grade	
POM	--	--	--	Yes	NR	15
POM/IFR	12.4	221.3	233.7	No	V-1	48.5
POM/IFR/1 wt%DPM	5	135.2	140.2	No	V-1	53.2
POM/IFR/2 wt%DPM	0	75.2	75.2	No	V-1	54.3
POM/IFR/3 wt%DPM	0	70.5	70.5	No	V-1	55.5
POM/IFR/4 wt%DPM	0	36.2	36.2	No	V-0	59.1
POM/IFR/5 wt%DPM	0	41.6	41.6	No	V-0	57.6
POM/IFR/6 wt%DPM	0	110.2	110.2	No	V-1	57.3
POM/IFR/7 wt%DPM	0	120.3	120.3	No	V-1	56.9
POM/IFR/8 wt%DPM	0	123.2	123.2	No	V-1	56.3

**Note:** The total additive amount of flame retardants and synergists in all systems was kept at 30 wt%, where the additive amount of DPM replaced the corresponding amount of IFR. t_1_ and t_2_ represent the total burning time of the first and second ignition, respectively; T_all_ is the total time of ten fires after two ignites.

**Table 3 polymers-16-00614-t003:** The main data of CONE test of composite materials.

Sample	POM	POM/IFR	POM/IFR/1 wt%DPM	POM/IFR/4 wt%DPM
TTI (S)	43	23	22	23
PkHRR (kW/m^2^)	335.55	163.17	132.79	129.17
PFI	7.8	7.1	6.0	5.6
AvHRR (kW/m^2^)	233.15	42.49	41.22	32.71
THR (MJ/m^2^)	133.08	89.47	95.45	64.90
MeanEHC (MJ/kg)	14.54	10.51	11.45	8.33
SEA (m^2^/kg)	0.00	86.30	159.40	109.31
AvMLR (g/(m^2^·s))	19.71	4.13	3.87	3.7
TSP (m^2^)	0.00	7.57	9.2	6.8
Residue (%)	0.0	11.3	15.0	15.5

**Table 4 polymers-16-00614-t004:** Thermogravimetric data of composite materials.

Sample	T_−5%_/°C	T_−10%_/°C	T_−50%_/°C	T_max_/°C	Actual Carbon Residue (600 °C)/%
POM	300.7	310.1	336.8	338.8	0.0
POM/IFR	261.7	263.1	271.5	269.1	17.0
POM/IFR/1 wt%DPM	260.1	261.2	271.1	266.8	21.8 (17.3 *)
POM/IFR/4 wt%DPM	262.6	263.9	274.1	268.8	23.5 (18.6 *)

**Note**: The data with “*” in brackets are the theoretical carbon residues of the sample, and its calculation formula is **Y = Y_POM/IFR_ × W_POM/IFR_ + Y_DPM_ × W_DPM_** (where “Y” represents the carbon residue amount and “W” represents the corresponding mass fraction).

**Table 5 polymers-16-00614-t005:** Mechanical data of composite materials.

Sample	Notched Impact Strength (kJ/m^2^)	Bending Modulus (MPa)	Bending Strength (MPa)	Tensile Strength (MPa)
POM	5.62 ± 0.06	2172.29 ± 22.57	70.37 ± 0.13	63.3.15
POM/IFR	2.78 ± 0.56	2707.04 ± 16.36	48.04 ± 0.34	36.20 ± 0.59
POM/IFR/1 wt%DPM	3.13 ± 0.21	2882.26 ± 17.23	48.25 ± 0.52	37.25 ± 0.32
POM/IFR/2 wt% DPM	3.25 ± 0.17	3058.51 ± 32.30	50.04 ± 0.63	37.87 ± 0.41
POM/IFR/3 wt% DPM	3.22 ± 0.05	2998.92 ± 42.26	50.32 ± 0.77	38.13 ± 0.33
POM/IFR/4 wt% DPM	3.28 ± 0.10	3067.59 ± 17.22	51.24 ± 0.11	38.68 ± 0.22
POM/IFR/5 wt% DPM	3.31 ± 0.08	3079.59 ± 25.21	52.39 ± 0.21	39.98 ± 0.15
POM/IFR/6 wt% DPM	3.32 ± 0.06	2968.26 ± 32.18	53.39 ± 0.54	40.03 ± 0.12
POM/IFR/7 wt% DPM	3.27 ± 0.12	3033.28 ± 32.09	53.44 ± 0.82	40.55 ± 0.52
POM/IFR/8 wt% DPM	3.28 ± 0.03	3200.23 ± 24.86	53.84 ± 1.85	41.00 ± 0.28

## Data Availability

Data are contained within the article.

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
