# Peer review of "Synergistic Modification of Polyformaldehyde by Biobased Calcium Magnesium Bi-Ionic Melamine Phytate with Intumescent Flame Retardant"

_polymers, 2024, doi:10.3390/polym16050614_

Round 1
Reviewer 1 Report
Comments and Suggestions for Authors
The article is dedicated to Synergistic Modification of Polyformaldehyde by Biobased Calcium Magnesium Bi-ionic Melamine Phytate with Intumescent Flame Retardant. It is written in good language and sounds scientifically. This manuscript is a whole study of the creation of a flame retardant containing polyformaldehyde with synergetic effect of DPM and can be interested for plastic manufacturers and specialists in the field.
Reviewer comments:
It seemed to reviewer that a lot of electronic cross-references are broken in the manuscript.
Page 2 line 3 “enterprises[2].At present” typo, space is missed.
First appearance of PLA on page 2 must be followed by its abbreviation decoding.
Page 3 Fig. 1. It contains typos in square brackets, upper right melamine ion must be cation as the rest three ones. Bottom right melamine cation must be in -NH3+ form. One instance of PA polyanion contains incorrect -OH- group. Total sum of negative and positive charges must be equal to zero.
Article contains two different references to the Fig. 1 which is confusing. Figures in manuscript must be renumbered.
Page 4 Chapter 3.1 “Figure 1.According” typo, space is missed.
Conclusion: Accept after minor revision.
Author Response
Dear Editors and Reviewers:
Thank you for your letter and the comments concerning our manuscript entitled " Synergistic Modification of Polyformaldehyde by Biobased Calcium Magnesium Bi-ionic Melamine Phytate with Intumescent Flame Retardant"(polymers-2820136).Those comments are valuable and helpful for us to improve the quality of our manuscript. We have carefully revised this manuscript and the revised parts have been highlighted in red. We appreciate your consideration for the publication of our revised manuscript. The responses to the reviewer’s comments or suggestions are listed as follows:
Reviewer #1:
1.It seemed to reviewer that a lot of electronic cross-references are broken in the manuscript.
2.Page 2 line 3 “enterprises[2].At present” typo, space is missed.
Response: Thanks very much. The formatting problems mentioned above in (1) and (2) have been corrected as suggested.
3.First appearance of PLA on page 2 must be followed by its abbreviation decoding.
Response: Thank you for your suggestion. I have added the full name of PLA in the position where it first appeared.
4.Page 3 Fig. 1. It contains typos in square brackets, upper right melamine ion must be cation as the rest three ones. Bottom right melamine cation must be in -NH3+ form. One instance of PA polyanion contains incorrect -OH group. Total sum of negative and positive charges must be equal to zero.
Response: Thank you for your advice. I'm sorry that due to my negligence, there are some mistakes in the writing of functional groups and ionic charges in Figure 1. I have corrected the mistakes in Figure 1 and uploaded a new picture.
5.Article contains two different references to the Fig. 1 which is confusing. Figures in manuscript must be renumbered.
6.Page 4 Chapter 3.1 “Figure 1.According” typo, space is missed.
Response: Thanks very much. The errors about (5) and (6) identified in the manuscript have been corrected.
Reviewer 2 Report
Comments and Suggestions for Authors
1. In the Abstract, the technical term ‘biobased calcium magnesium bi-ionic melamine phytate’ was abbreviated as DPM. It is noteworthy that the first two letters of this abbreviation do not correspond to any words in the original term. Also, the term ‘biobased’ can be omitted, as phytic acid is inherently derived from natural sources. The inclusion of this term does not promote the value of the studied composite.
2. Additionally, it is important to note that this manuscript does not reference Wei Yang’s work (https://doi.org/10.1016/j.compositesa.2018.04.027) concerning the synthesis of this specific fire retardant, calcium magnesium melamine phytate. However, it does cite another study by the same researcher, focusing on a different fire retardant, i.e., phytic acid and hexakis (4-aminophenoxy) cyclotriphosphazene.
3. In the Abstract, it is advisable to consistently use either ‘rating’ or ‘grade’ to avoid misunderstandings. Moreover, the frequent use of abbreviations throughout the paper can be confusing; the authors should address this issue.
4. In the first paragraph of the Introduction, the terms of ‘IFR’ and ‘matrix’ are introduced without specifying a candidate substance. The relationship of POM with IFR is not clear. Consequently, the claim that “IFR has poor compatibility with the matrix” lacks clarity and requires further explanation.
5. In the second paragraph of the Introduction, there is a lack of a clear depiction of the ideal polymer composite structure/composition to minimize factors such as, LOI, smoke volume, combustible volatile, etc. The logic behind these ignitable properties is also not well defined.
6. The last illustration in Figure 1 contains a minor error in one of the bonding arms between phosphate and melamine. Importantly, the chemical structure portrayed in this drawing is merely a hypothesis, given that the preparation method outlined in section 2.1 does not involve a homogeneous solution reaction system. The resulting composite is, in fact, a heterogeneous system with a spatial composition distribution at a dozen micron scale, as evidenced by the SEM image in Figure 3.
7. In Figure 5b, there are inaccuracies in the assignments of the IR absorption bands, specifically, the band at about 2350 cm-1 shouldn’t be attributed to CºO, and similarly, the band at about 1600 cm-1 should not be assigned to S=O. Additionally, Figure 5b should include the TG curve of curves of POM and IFR, not just the TG curve of DPM. It is crucial to note that the purging gas during TGA should be also a mixture of N2 and O2 in a specific proportion.
8. Regarding the statement in Section 3.6, ‘Ca2+ and Mg2+ in DPM can not only catalyze the esterification reaction between acid and carbon source…’, it is presented as speculation. The authors should, at least, provide a reference or a supporting evidence for this claim.
Comments on the Quality of English Language
The quality of English is reasonably satisfactory.
Author Response
Dear Editors and Reviewers:
Thank you for your letter and the comments concerning our manuscript entitled " Synergistic Modification of Polyformaldehyde by Biobased Calcium Magnesium Bi-ionic Melamine Phytate with Intumescent Flame Retardant"(polymers-2820136).Those comments are valuable and helpful for us to improve the quality of our manuscript. We have carefully revised this manuscript and the revised parts have been highlighted in red. We appreciate your consideration for the publication of our revised manuscript. The responses to the reviewer’s comments or suggestions are listed as follows:
1.In the Abstract, the technical term ‘biobased calcium magnesium bi-ionic melamine phytate’ was abbreviated as DPM. It is noteworthy that the first two letters of this abbreviation do not correspond to any words in the original term. Also, the term ‘biobased’ can be omitted, as phytic acid is inherently derived from natural sources. The inclusion of this term does not promote the value of the studied composite.
Response: Thank you for your question. First, the abbreviation “DPM” is based on the three main raw materials of DDS (4, 4-diaminodiphenyl sulfone), PA (phytic acid) and ME (melamine). Second, low smoke, low toxicity, environmentally friendly flame retardants are the future development trend. The term ‘biobased’ used here aims to further emphasize the environmentally friendly properties of phytate flame retardants.
2.Additionally, it is important to note that this manuscript does not reference Wei Yang’s work (https://doi.org/10.1016/j.compositesa.2018.04.027) concerning the synthesis of this specific fire retardant, calcium magnesium melamine phytate. However, it does cite another study by the same researcher, focusing on a different fire retardant, i.e., phytic acid and hexakis (4-aminophenoxy) cyclotriphosphazene.
Response: Thank you for your suggestion, and I am very sorry that there are errors in the references in the article. Wenxue Yang’s work on phytic acid and six (4-aminophenoxy) cyclotriphosphonitrile flame retardants (reference 18) was cited in the second paragraph of the introduction. In Section 3.1, Wei Yang 's method of synthesis and characterization of melamine phytate calcium magnesium flame retardants (reference 23) was also cited.
3.In the Abstract, it is advisable to consistently use either ‘rating’ or ‘grade’ to avoid misunderstandings. Moreover, the frequent use of abbreviations throughout the paper can be confusing; the authors should address this issue.
Response: Thank you very much. It has been modified.
4.In the first paragraph of the Introduction, the terms of “IFR” and “matrix” are introduced without specifying a candidate substance. The relationship of POM with IFR is not clear. Consequently, the claim that “IFR” has poor compatibility with the “matrix” lacks clarity and requires further explanation.
Response: Thank you for your suggestion. In this paper, we have modified “IFR” to “IFR additives” and "matrix" to "polymer matrix" in order to better distinguish the relationship between matrix and IFR. In this study, the prepared DPM and IFR are both modified plastic additives, which are added to the polymer matrix to achieve flame retardant modification of the polymer.
5.In the second paragraph of the Introduction, there is a lack of a clear depiction of the ideal polymer composite structure/composition to minimize factors such as, LOI, smoke volume, combustible volatile, etc. The logic behind these ignitable properties is also not well defined.
Response: Thank you for your comments. The second paragraph of the introduction mainly summarizes some means to improve the flame-retardant efficiency and the compatibilityof the polymer and additives, and briefly lists the relevant work of others. In the section of "Results and Discussion", the composition of the composite material and the relevant data such as LOI, heat release and smoke release are explained and analyzed in detail.
6.The last illustration in Figure 1 contains a minor error in one of the bonding arms between phosphate and melamine. Importantly, the chemical structure portrayed in this drawing is merely a hypothesis, given that the preparation method outlined in section 2.1 does not involve a homogeneous solution reaction system. The resulting composite is, in fact, a heterogeneous system with a spatial composition distribution at a dozen micron scale, as evidenced by the SEM image in Figure 3.
Response: Thanks for your comments. As you said, the chemical structure portrayed in this Figure 1 is indeed a hypothesis. The error in Figure 1 has been corrected.
7.In Figure 5b, there are inaccuracies in the assignments of the IR absorption bands, specifically, the band at about 2350 cm-1 shouldn’t be attributed to C=O, and similarly, the band at about 1600 cm-1 should not be assigned to S=O. Additionally, Figure 5b should include the TG curve of curves of POM and IFR, not just the TG curve of DPM. It is crucial to note that the purging gas during TGA should be also a mixture of N2 and O2 in a specific proportion.
Response: Thank you very much for your suggestion. The purpose of only testing DPM in Figure 5 is to analyze the thermal stability of DPM and the composition of the gas released when fully burned in a nitrogen atmosphere. According to the literature, the most significant peak position of CO2 in the infrared spectrum is around 2349 cm-1, and the double bond stretching vibration absorption peak of S=O often appears near the band of 900~1800cm-1. According to a series of test analysis and speculation, the bands around 2350 cm-1 and 1600 cm-1 should probably belong to CO2 and S=O. Additionally, the TG/DTG curve of POM, POM/IFR and POM/IFR/DPM composite were included in section 3.5 (Figure8) .
8.Regarding the statement in Section 3.6, “Ca2+” and “Mg2+” in DPM can not only catalyze the esterification reaction between acid and carbon source…’, it is presented as speculation. The authors should, at least, provide a reference or a supporting evidence for this claim.
Response: Thank you very much, “metal ions can catalyze the esterification cross-linking reaction between acid source and carbon source, and the rapid carbonization on the collective surface of the polymer” this conclusion has been confirmed in IFR flame retardant mechanism studies and several reports. I have added a reference for this claim.
Reviewer 3 Report
Comments and Suggestions for Authors
The work is devoted to study of role of calcium magnesium bi-ionic melamine phytate (DPM) additive to flame retardant polyformaldehyde (POM) composite. Based on the comprehensive investigation of fire retardant properties of the obtained samples (including various types of analysis) the authors clearly demonstrated the synergistic role of DPM and proposed the flame retarding mechanism.
The methods are clearly described, the results are described in details. However, it would be nice to find the comparison of the obtained results with literature data in discussion part. Moreover, the results of all provided analysis of all obtained samples must be presented in supplementary materials. Despite my critics the present work is of very high quality and deserves to be published in Polymers. Below you can find some of questions/corrections.
1) I am not sure about the correctness of the phrases "environmental friendliness", "addition amount"
2) Speaking about EDS, what is the detection limit of your detector? What is the detector model?
3) The justification of DPM formula/content is not 100% reliable. The IR bands of DDS and ME overlap in region of 3500-3200 cm-1, the same is observed for PA, ME and PA in 1750-1500 cm-1 region. In order to obtain reliable quantitative results of the element composition with help of SEM-EDS the sample should be polished. In your case the sample was a powder, hence there is no ground to claim the composition. To provide more reliable chemical composition, please, employ XRF.
4) The Figure 2 is absent.
5) The legend of Figure 1 is too short, please, provide more detailed description.7) Figures 5 precedes Figure 4 in text. It should be the other way around.
6) Only Table 1 contains information on all samples obtained in the study. In the next tests the authors provide the comparison of pure POM, POM/IFR, POM/IFR/1wt%DPM and POM/IFR/4wt%DPM. I guess that introduction of the rest of samples to the next tables in figures will make them too overloaded. Nevertheless it would nice to include this information in supplementary materials, since it would be interesting, for example, to see how the amount of DPM affects the char residue? Could you provide the corresponding data from all analysis for the rest of the samples?
7) Considering the LOI and rating in Table 1 why there is an optimum around 4-6%wt of DPM?
8) The quality of Figure 9 is not good enough to read clearly the information in it. Could you provide the better quality pictures, please?
9) "Combined with the cone calorimetric analysis of each system and the carbon residue analysis, it can be seen that the basic C-P, P=O skeleton of phytic acid and metal ions in the combustion process of DPM mainly play a catalytic role in the condensed phase leading to rapid carbonization." - is it in agreement with literature data? Could provide examples, please?
Author Response
Dear Editors and Reviewers:
Thank you for your letter and the comments concerning our manuscript entitled " Synergistic Modification of Polyformaldehyde by Biobased Calcium Magnesium Bi-ionic Melamine Phytate with Intumescent Flame Retardant"(polymers-2820136).Those comments are valuable and helpful for us to improve the quality of our manuscript. We have carefully revised this manuscript and the revised parts have been highlighted in red. We appreciate your consideration for the publication of our revised manuscript. The responses to the reviewer’s comments or suggestions are listed as follows:
1.I am not sure about the correctness of the phrases “environmental friendliness”, “addition amount”.
Response: Thank you for your reminding. I have changed them to “environmental friendly” and “additive amount” respectively.
2.Speaking about EDS, what is the detection limit of your detector? What is the detector model?
Response: Thanks for your suggestion, I have added the detection limits and detection model of EDS test to the corresponding places in the manuscript.
3.The justification of DPM formula/content is not 100% reliable. The IR bands of DDS and ME overlap in region of 3500-3200 cm-1, the same is observed for PA, ME and PA in 1750-1500 cm-1 region. In order to obtain reliable quantitative results of the element composition with help of SEM-EDS the sample should be polished. In your case the sample was a powder, hence there is no ground to claim the composition. To provide more reliable chemical composition, please, employ XRF.
Response: Thank you very much for your suggestion. The chemical structure of DPM portrayed in Figure 1 is merely a hypothesis. DPM is, in fact, a heterogeneous system with a spatial composition distribution at a dozen micron scale, as evidenced by the SEM image in Figure 3. And it is insoluble in water and many organic solvents. Therefore, only the method described in the article was used for preliminary characterization.
4.The Figure 2 is absent.
5.The legend of Figure 1 is too short, please, provide more detailed description. Figures 5 precedes Figure 4 in text. It should be the other way around.
Response: Thank you very much for your suggestion. The two problems of 4 and 5 have been corrected in the manuscript.
6.Only Table 1 contains information on all samples obtained in the study. In the next tests the authors provide the comparison of pure POM, POM/IFR, POM/IFR/1wt%DPM and POM/IFR/4wt%DPM. I guess that introduction of the rest of samples to the next tables in figures will make them too overloaded. Nevertheless it would nice to include this information in supplementary materials, since it would be interesting, for example, to see how the amount of DPM affects the char residue? Could you provide the corresponding data from all analysis for the rest of the samples?
Response: Thank you for your comments. In this study, only a few representative composite samples (pure POM, POM/IFR, POM/IFR/1wt%DPM and POM/IFR/4wt%DPM) were selected for subsequent series of test analysis. As it should be, we can provide the corresponding data from all analysis for the rest of the samples. However, we think there are enough data to support the conclusion of this study.
7.Considering the LOI and rating in Table 1 why there is an optimum around 4-6%wt of DPM?
Response: Thank you for your question. From the results of LOI and UL94 ratings in Table 1, it can be seen that with the increase of DPM introduction, the LOI of the composite material presents a trend of first rising and then decreasing, and the combustion time also presents a phenomenon of first decreasing and then increasing, and the optimal flame-retardant effect is achieved when 4wt%DPM is introduced. The LOI of POM composites added with 4wt% ~6wt%DPM are all above 57%, and at the same time they can reach UL94 V-1 level or above. So 4wt% ~6wt% is an optimum additive amount of DPM.
8.The quality of Figure 9 is not good enough to read clearly the information in it. Could you provide the better quality pictures, please?
Response: Thanks to your suggestion, I have replaced the higher definition photos in the manuscript.
9.“Combined with the cone calorimetric analysis of each system and the carbon residue analysis, it can be seen that the basic C-P, P=O skeleton of phytic acid and metal ions in the combustion process of DPM mainly play a catalytic role in the condensed phase leading to rapid carbonization.” - is it in agreement with literature data? Could provide examples, please?
Response: Thank you very much for your comments. This inference is consistent with the flame retardant mechanism of phytic acid metal salts reported in literature, such as reference 23. It also agrees with the results of TG-IR, cone calorimetry and post-combustion carbon residue analysis. DPM has both gas phase and condensed phase flame retardancy. N and S element in DPM mainly play a role in diluting combustible gas in the gas phase, while P element of C-P, P=O skeleton and metal ions mainly play a flame-retardant role in the condensed phase through catalyzing the rapid carbonization of the flame-retardant polymer composites.
Round 2
Reviewer 3 Report
Comments and Suggestions for Authors
The authors have followed some of my advises and made the corresponding correction. However, they absolutely ignored another part of me quires. So, I have to go back to them again. Here they are:
- The authors provided point elemental analysis as they wrote in experimental part. Please, say how many points have been measured? Please, provide the corresponding errors for the elemental composition, e.g. O - 17.19(0.20), etc. I still insit on XRF, since the EDS measurements provided in this work are really far from the reliable data.
- Please, provide the data from all analysis for the rest of the samples in Supplementary materials.
- Unfortunately I cannot accept the author's reply on my question #7. The description of the results is not the explanation of them! I still need a clear explanation why one can see the best values of LOI and rating in Table I in case of samples with 4-6%wt of DPM. Please, do the necessary explanation in the text of manuscript.
- Unfortunately the authors ignored my recommendation to provide the literature references that can support thier statement: "Combined with the cone calorimetric analysis of each system and the carbon residue analysis, it can be seen that the basic C-P, P=O skeleton of phytic acid and metal ions in the combustion process of DPM mainly play a catalytic role in the condensed phase leading to rapid carbonization". Please, describe in the manuscript whether your results are in agreement with literature data or not? Compare the mechanism of the flame retardance proposed previously with yours (also should be included in the discussion part). Provide more than 1 reference.
Author Response
Dear Editors and Reviewers:
Thank you for your letter and the comments concerning our manuscript entitled " Synergistic Modification of Polyformaldehyde by Biobased Calcium Magnesium Bi-ionic Melamine Phytate with Intumescent Flame Retardant"(polymers-2820136).Those comments are valuable and helpful for us to improve the quality of our manuscript. We have carefully revised this manuscript and the revised parts have been highlighted in red. We appreciate your consideration for the publication of our revised manuscript. The responses to the reviewer’s comments or suggestions are listed as follows:
- The authors provided point elemental analysis as they wrote in experimental part. Please, say how many points have been measured? Please, provide the corresponding errors for the elemental composition, e.g. O - 17.19(0.20), etc. I still insist on XRF, since the EDS measurements provided in this work are really far from the reliable data.
Response: Thank you for your patient guidance and suggestions. First of all, the data in Figure 3 were measured by using the point analysis model to test the selected range as a whole and taking the average value, instead of selecting multiple points for separate testing. According to your suggestion, we added tests on multiple points, and the results are shown in the table below. However, point analysis has a large error and is not suitable for characterizing the element content of the substance, so we did not choose to put this data in the manuscript. In addition, EDS mapping analysis (element-oriented image distribution) was added to the manuscript. As shown in the scan diagram in this manuscript, the test is a selected range, and the test result is also the average value within the range, so that the distribution of each element can be seen more intuitively, and the measured data is closer to the theoretical value. Based on your suggestions, we performed the EDS mapping analysis in Figure 3 and modified the corresponding data.
Moreover, according to your suggestion, we have supplemented the XRF test analysis in the "3.1 Characterization of DPM" and the specific parameters and models of the X-ray fluorescence spectrometer in the "2.3 Characterization "in the manuscript.
- Please, provide the data from all analysis for the rest of the samples in Supplementary materials.
Response: Thank you for your suggestion. But we are very sorry that we cannot provide all the analysis data for the rest of the samples at the moment. The reasons are as follows: 1) Based on the test results in Table 2, we selected only four representative samples (pure POM, POM/IFR, POM/IFR/1 wt%DPM and POM/IFR/4 wt%DPM) for subsequent testing and further exploration. And the final conclusion is drawn by comparison. 2) If supplement the data of the rest samples, we need more time to do that. We don't think it's very necessary. Due to the small variation in the amount of DPM (1wt%) added to the samples, there should show a certain trend and have little change in the residual carbon, heat release, thermal analysis, etc. according to our past experience. Therefore, we are awfully sorry for not being able to provide the corresponding data. But we add and explain the reasons for the sample selection in the section 3.2 of the manuscript.
- Unfortunately I cannot accept the author's reply on my question #7. The description of the results is not the explanation of them! I still need a clear explanation why one can see the best values of LOI and rating in Table I in case of samples with 4-6%wt of DPM. Please, do the necessary explanation in the text of manuscript.
Response: Thank you for your question, and I am sorry that I did not fully understand your meaning last time. On this occasion, we have supplemented some explanation in the section 3.2 of the manuscript about how to judge the flame-retardant grade and the definition of LOI. Thus, it can be seen that the shorter the total burning time (Tall), the more difficult the material is to ignite. And the higher the LOI, the better the flame retardant performance. Moreover, the definition of t1, t2 and Tall were added in the "Note" after Table 2 in the paper.
4.Unfortunately the authors ignored my recommendation to provide the literature references that can support their statement: "Combined with the cone calorimetric analysis of each system and the carbon residue analysis, it can be seen that the basic C-P, P=O skeleton of phytic acid and metal ions in the combustion process of DPM mainly play a catalytic role in the condensed phase leading to rapid carbonization". Please, describe in the manuscript whether your results are in agreement with literature data or not? Compare the mechanism of the flame retardance proposed previously with yours (also should be included in the discussion part). Provide more than 1 reference.
Response: Thank you very much. Additional references have been made in the sections "3.5 Thermogravimetric - infrared analysis" and "3.6 Flame retardant mechanism analysis" according to your suggestion. The sentence of "Combined with the cone calorimetric analysis of each system and the carbon residue analysis, it can be seen that the basic C-P, P=O skeleton of phytic acid and metal ions in the combustion process of DPM mainly play a catalytic role in the condensed phase leading to rapid carbonization" was revised in the manuscript to “Combined with the cone calorimetric analysis of each system and the carbon residue analysis, it can be seen that mutual reactions occurred among DPM, IFR and POM in the combustion process. DPM mainly play a catalytic role in the condensed phase leading to rapid carbonization and more high-quality char residue [23,27].” In the sections of "3.6 Flame retardant mechanism analysis", we also point out that the mechanism of IFR/DPM proposed is consistent with the action mechanism of phytate reported in the previous literature.
Thanks again for all your suggestions!

Round 3
Reviewer 3 Report
Comments and Suggestions for Authors
The manuscript in its present form can be published in Polymers.